# Robot-Assisted Gait Training with Trexo Home: Users, Usage and Initial Impacts

**DOI:** 10.3390/children10030437

**Published:** 2023-02-24

**Authors:** Christa M. Diot, Jessica L. Youngblood, Anya H. Friesen, Tammy Wong, Tyler A. Santos, Benjamin M. Norman, Kelly A. Larkin-Kaiser, Elizabeth G. Condliffe

**Affiliations:** 1Alberta Children’s Hospital, Calgary, AB T3B 6A8, Canada; 2Cumming School of Medicine, University of Calgary, Calgary, AB T2N 4N1, Canada; 3Schulich School of Engineering, University of Calgary, Calgary, AB T2N 4N1, Canada

**Keywords:** robotics, cerebral palsy, neurological rehabilitation, sleep, bowel function

## Abstract

Robotic gait training has the potential to improve secondary health conditions for people with severe neurological impairment. The purpose of this study was to describe who is using the Trexo robotic gait trainer, how much training is achieved in the home and community, and what impacts are observed after the initial month of use. In this prospective observational single-cohort study, parent-reported questionnaires were collected pre- and post-training. Of the 70 participants, the median age was 7 years (range 2 to 24), 83% had CP, and 95% did not walk for mobility. Users trained 2–5 times/week. After the initial month, families reported a significant reduction in sleep disturbance (*p* = 0.0066). Changes in bowel function, positive affect, and physical activity were not statistically significant. These findings suggest that families with children who have significant mobility impairments can use a robotic gait trainer frequently in a community setting and that sleep significantly improves within the first month of use. This intervention holds promise as a novel strategy to impact multi-modal impairments for this population. Future work should include an experimental study design over a longer training period to begin to understand the relationship between training volume and its full potential.

## 1. Introduction 

For people with significant mobility limitations, many aspects of health are impacted. Physical inactivity, poor sleep, low mood, and constipation are often reported as secondary complications of limited mobility. People living with disabilities are up to 62% less likely to meet recommended physical activity guidelines than people without disabilities. For children with a disability, less than 10% meet physical activity guidelines [1]. Children with cerebral palsy (CP), the most common cause of physical disability in children, have a higher incidence of sleep-related problems than children in the general population [2,3], and 55% of children with severe neurologic impairment report insomnia [4]. Further, non-ambulatory children experience greater severity of sleep problems than ambulatory children with CP [5]. Inadequate sleep in children is also related to decreased sleep quality for caregivers, which impacts parent/caregiver psychological health and well-being [5,6,7]. Constipation is also more common in children with CP [8] and depending on how it is defined, it is prevalent in up to 74% of children with CP [9,10]. Increasing physical activity can yield health benefits in the previously described domains in addition to the other long-term sequalae of inactivity for people living with a disability [1].

Over the past decade, robot-assisted gait trainers (RAGT) have gained interest in rehabilitation settings for people with mobility limitations. There are now many commercially available RAGT options. Of the few options suitable for children, children with the most severe mobility limitations are rarely included in the literature, which often focuses on gait outcomes [11,12,13]. Ultimately, who may benefit from using a RAGT is unknown. Arguably, exploring the impacts on the most impaired children could be even more meaningful since options for upright assisted movement are scarce, particularly if the focus shifts from rehabilitation of gait to exploring broader impacts of assisted mobility, including secondary health conditions. There is some evidence of improvements in secondary health conditions for adults post-stroke or spinal cord injury with the use of RAGT [14,15,16,17]. With most of the evidence related to RAGT use coming from the sub-acute clinical rehabilitation setting, the practical and feasible volume of use in a community setting has not been determined. It is also unclear what changes might be observed with initial onset of RAGT use for children with severe mobility limitations, as in some cases, RAGT are the only way to experience walking. There is some evidence of training benefits with RAGT use in children with CP [12,18,19], and a case study involving a significantly impaired child found improvements in bowel function and head control, as well as reduced knee flexor spasticity and anecdotal reports of improved sleep [20]. A stationary robotic trainer has demonstrated the potential for provision of physical activity in children [15]. It seems reasonable to consider that RAGT use may provide benefits associated with reducing sedentary time for children with significant mobility limitations.

Trexo Robotics (Mississauga, ON, Canada) has developed an over ground robotic gait trainer for children that can be used at home, school, and in the community, called the Trexo Home (Trexo) (Figure 1). The robotic legs have an external power source and the unit attaches to a stable four wheeled walker, allowing for safe mobilization on a regular basis in the community. The flexibility of using such a device in home, school, and community settings is novel and allows patients and families to dictate training volume without needing costly clinic and therapist time. This exploratory observational study gathers information from new Trexo users to help us understand who might benefit from using the device and what volume of use is practical and feasible for families. Further, we test hypotheses that the initial month of RAGT use impacts secondary health conditions associated with immobility that are highly prevalent in children with mobility limitations.

## 2. Methods

### 2.1. Design

The study was approved by the Conjoint Health Research Ethics Board (REB20-0799). Participants who had already leased or purchased a Trexo were invited to participate in this prospective observational single-cohort study. Volume and timing of Trexo use was dictated by the participants’ families based on what was feasibly achieved in the community for each participant.

### 2.2. Setting

Participants used the Trexo in their home, school, and/or community, including indoors and outdoors. The electronic nature of data collection allowed participants to live anywhere in the world with internet access; however, most participants lived in Canada and the United States.

### 2.3. Participants

Participants were eligible for inclusion if they committed to obtain a Trexo (either leased or purchased independent of this study) and provided informed consent via electronic signature.

### 2.4. Outcome Measures

Demographics and parent-proxy outcomes were collected via electronic surveys (Jotform) sent by Trexo Robotics with follow-up communication to facilitate completion by the research team and/or Trexo Robotics.

#### 2.4.1. Participant Demographics

Parents were asked information about their child’s age, diagnosis, and functional ability. Functional ability was characterized by three scales: Gross Motor Function Classification System—Expanded & Revised (GMFCS-E&R), the Gillette Functional Ability Questionnaire: Functional Walking Scale (FAQ-10), and the Functional Mobility Scale (FMS). The Gross Motor Function Classification System (GMFCS) Family Report Questionnaire [21,22] is a validated tool used to help classify children and youth with physical disabilities into five functional categories ranging from I (walks without limitations) to V (transported by others in a manual wheelchair). The FAQ-10 is a 10-level parent report of walking ability considering external factors such as stepping with support, where 1 represents no stepping ability and 10 indicates independent walking and climbing on varied terrain without difficulty [23]. The FMS is used to describe functional mobility over three distinct distances (5, 50, and 500 m), representing home, school, and community settings with each distance ranked 1 (uses wheelchair) to 6 (independent on all surfaces) with additional categories C (crawling) and N (does not complete this distance on own) [24].

#### 2.4.2. Usage Data

Information from each session using the Trexo was stored on a tablet that connected to the Trexo device via Wi-Fi and was sent automatically to Trexo Robotics at the end of a training session. This information included the date and start/end time of each use, duration of use, and number of steps taken. Trexo Robotics gave the research team access to their database for participants in this study.

Calculations for intensity of use were based on 7-day epochs, where day one was the date of initial training to use the Trexo, and each month was calculated as 28 days to maintain consistency when looking at the initial month of use with participants who had varied start dates. Weekly and monthly averages were calculated for number of sessions and steps.

#### 2.4.3. Parent-Proxy Outcomes

Patient-reported outcomes through parent proxies were used to evaluate the impacts of one month of training. Specifically, items related to prevalent secondary conditions potentially linked to physical activity and a measure of physical activity were selected. Parents reported on their child’s bowel function by recalling details from the past week. Frequency was selected as one of five options (more than daily; daily; every other day; two times in the last week; less than two in the last week). Parents were also asked to report any medication, procedures, or dietary strategies associated with bowel function as free text. Participants were subcategorized as ‘constipated’ if they had a bowel movement frequency of 2×/week or less or if they used any constipation-related medications or procedures.

The Patient-Reported Outcome Measurement Information System (PROMIS^®^ National Institutes of Health, Bethesda, MD, USA) was used to document sleep disturbance, positive affect, and physical activity. PROMIS^®^ is a set of validated measures that evaluate physical, mental, and social health in adults and children, including people living with chronic conditions. For all PROMIS^®^ measures used, the questions were based on a 7-day recall period. All scores were converted to a t-score with 50 as the mean of the general population. Parent proxy measures were designed and validated for parents to report on their child from age 5–17, and the same forms were used for all participants in this study. Participants had to use the Trexo, operationally defined as >100 steps within the first month, for their outcomes to be evaluated.

Sleep disturbance was reported using the PROMIS^®^ Parent Proxy (Item Bank v1.0–Sleep Disturbance—Short Form 8a) [25], which assesses perceptions around sleep quality, as well as falling asleep and staying asleep. Severity of sleep disturbance was categorized such that a *t*-score of 56–59 is considered mild, 60–65 was moderate, and ≥66 was severe. A *t*-score ≤ 55 was considered within normal limits [26,27].

Positive affect was reported via the PROMIS^®^ Parent Proxy (Item Bank v1.0–Positive Affect—Short Form 8a) [28], which assesses a child’s positive or rewarding experiences, including pleasure, joy, pride, and happiness.

Physical activity was reported using the PROMIS^®^ Parent Proxy (Item Bank v1.0—Physical Activity—Short Form 4a) [29], which provides a valid measurement of children’s lived experiences of physical activity in the general population.

### 2.5. Timing of Assessments

All outcome measures were to be collected 1–2 times prior to onset of Trexo use, though some participants completed initial assessments during the first few weeks of use. All outcome measures were collected again after using the Trexo for 4–8 weeks. The PROMIS^®^ Physical Activity questionnaire was added after two participants had already completed a baseline assessment, so there are two fewer respondents for this specific outcome measure.

### 2.6. Statistical Methods

Baseline assessments completed within three months of starting with the Trexo and within the first two weeks of Trexo use were included. Baseline assessments were excluded if completed outside of the specified time frame. In cases where participants completed two baseline assessments within the specified time frame, the average of the two baselines was used. Participants who did not have a completed baseline assessment or did not have a completed 1-month assessment were excluded from the outcomes analysis but were included in the participant demographics and usage data analysis.

Non-parametric descriptive statistics are presented as the median (25th–75th percentiles). Statistical differences in parent proxy outcomes were tested with the Wilcoxon signed-rank test with significant differences detected at *p* < 0.0125, (α = 0.05 with Bonferroni correction for 4 independent tests).

## 3. Results

Over the course of 15 months, 116 Trexo units were deployed to potential participants. Participant flow is outlined in Figure 2 and involved 109 families who provided consent to be contacted about recruitment, with 70 providing informed consent. Two participants were excluded due to contraindication to use (one had a medical contraindication and one did not fit the size requirements).

Age, sex, and diagnosis were provided by all 70 participating families. Of the 70 participants with signed consent, usage data were included for 66 participants, with two excluded due to technical issues and another two excluded because they were using the same device and individual usage data were unable to be differentiated. Outcome data from parent proxy questionnaires were included for 42 participants, the other 28 were excluded. Among them, 26 participants had incomplete data, one was the participant unable to use the Trexo and so was also excluded from the usage data, and one had fewer than 100 steps total in the initial month, which we feel is too low to inform the impact of usage. The PROMIS^®^ Physical Activity outcomes were added after the first two participants had already completed baseline assessments, so physical activity data were included for 40 participants. 

### 3.1. Participants

The participants were predominantly elementary school age, non-ambulatory children with CP (Table 1). Of the 70 participants with demographic information, the median (25th–75th percentile) age was seven (four to nine) years and 41 participants (59%) were male. Among 63 (95%) participants with parent-reported GMFCS, 60 were classified as levels IV or V, indicating that most participants required significant physical assistance and had limited independent mobility. A total of 58 participants (83%) had a non-progressive diagnosis, five participants had a progressive diagnosis, and seven did not have a confirmed diagnosis or had a diagnosis for which potential long-term progression was unknown.

The sample population experienced significant mobility limitations (Figure 3). Baseline FAQ-10 scores were reported by 49 participants, and of those, 80% reported a score of 1 or 2, meaning they may take some supported steps but do not walk on a routine basis. Only three participants reported an FAQ-10 score of ≥6, meaning they could walk at least short distances outside on level ground. Most participants were not able to complete any distance independently, as reported in the FMS. 

### 3.2. Trexo Usage

The volume of Trexo use was highly variable in the initial month, with many participants having large fluctuations in usage from week to week. Of the 66 participants included in the usage data analysis, median (25th–25th percentile) usage was 732 (284–1795) steps/week split over usage three (two to five) times. A few participants’ usage was far beyond this distribution, with ranges in steps/week from 21 to 23,170. During the first month, 5 participants reported a broken or malfunctioning Trexo component, while another 5 participants had missing data as a result of internet connectivity issues. Four participants reported periods of non-use for personal reasons such as travel or moving houses.

### 3.3. Parent-Proxy Outcomes

Bowel function—Of the 42 participants who reported bowel function, 21 were constipated (3/21 defecating ≤2×/week and 20/21 taking medication). Despite this high prevalence of constipation, the median change in bowel movement was 0 (−1–+1), indicating no change over the first month of use (*p* = 1.0).

Sleep disturbance—At baseline, 93% of participants had a t-score above 50, meaning they experienced more sleep disturbance than the reference population, with a median score of 63.9 (55.8–66.2) (Figure 4). Sleep disturbance was categorized as clinically meaningful for 76% of participants. After one month of using the Trexo, there was a statistically significant reduction in sleep disturbance by −2.4 (−5.6–0.7), *p* = 0.0066 (Figure 5). Of the 32 participants with clinically meaningful sleep disturbance, 15 (74%) either no longer had sleep disturbances or the severity of sleep disturbance was classified in a less severe category. The number of participants with a classification of severe sleep disturbance was 13/42 at baseline and 9/42 after the initial month of Trexo use.

Positive affect—The positive affect scores were slightly below the reference distribution with a median of 45.5 (37.5–49.8) at baseline (Figure 4). While there was an increase in positive affect after one month of Trexo use to 48.1 (42.5–52.1), with a median change of 2.65 (0.0–6.1) (Figure 5), it did not represent a significant increase (*p* = 0.0129). 

Physical activity—The sample population reported physical activity levels below the mean of the general population. The median (25th–75th percentile) physical activity score at baseline was 42.5 (38.1–46.5) (Figure 4) and the one-month scores had a median of 45.6 (37.7–48.2). The median change in physical activity scores was 1.2 (−5.1–5.8) following one month of Trexo use, as seen in Figure 5 (*p* = 0.556).

## 4. Discussion

This RAGT, designed for community use without a trained professional supervising each session, is new and novel (available since 2019). This observational study provides the first description of its users, usage, and initial impacts. While the users’ age and size ranges were broad, from a toddler to a skeletally mature young adult, the commonality among most users was their primary use of a wheelchair for mobility. Within the first month with the Trexo, participants took at least 300 steps/week with most taking more than double that number. Concurrently, participants began sleeping better. While the activity change caused by taking steps seems to be the probable cause of this improvement, our assessment of physical activity levels did not detect any change.

Removing the need for a clinical setting or clinicians to use a RAGT reduces two large barriers to frequent training. Training 2–5×/week is typically more frequent than is clinically available. There was tremendous variability in usage volume over the initial month. Many factors contributed to this, including the fact that data were collected during the global COVID-19 pandemic which impacted supply chains and created shipping delays. This was commonly reported as a reason for downtime when replacement parts were required. It also made RAGT arrival time more variable and limited families’ abilities to prepare for this new addition to their routines. It is also possible that there was a familiarization period for the family and participant during the initial month. 

Studying patient-reported outcomes related to novel technologies in an observational study enables the evaluation of real-world use and inclusion of larger populations than are typically feasible in single-centre experimental studies. A reduction in sleep disturbances was robustly reported. A minimally important difference has not been established for this outcome; however, the median magnitude (2.4) was around the minimally important difference of 2–3 that has been determined for multiple other PROMIS outcomes [30,31]. Further, the sleep disturbances in almost 1/3rd of those categorized as severe at baseline were no longer severe after 1 month of training with the RAGT and almost half of those with clinically meaningful sleep disturbances moved to a less severe category or no longer had sleep disturbances. These reductions may have been caused by greater levels of physical activity, reduced pain, reduced involuntary movement or maintenance of posture (caused by spasticity, dystonia, or athetosis), or improved mood, but further research is required to elucidate the cause. 

Despite positive comments about bowel habits, affect and physical activity levels from parents and some trends toward improvements in the results, we did not detect significant changes in these outcomes. Interestingly, the median increase in positive affect was of similar magnitude to the improvement in sleep, but the change was more variable and it did not reach our conservative threshold for a significant difference. While this study used validated outcomes, we may not have selected the ideal outcomes for this specific population. All but one of our constipated participants used medication to support their bowel function. It is possible that training with the RAGT does improve bowel function, but that this is reflected by changes in stool quality or medication requirements and not consistently in frequency, despite finding more frequent bowel movements in a previous case study [20]. Some parents anecdotally reported improvements in positive affect and physical activity but found this was not captured well by the questions included in the respective item banks due to the differences in their child from typical developing children. For example, one question asked about sweating, which parents reported was a poor indicator of physical activity in their children with neurologic impairment, an observation that is supported by our clinical experience and some evidence [32]. Greater patient engagement in the selection of our patient-reported measures and possibly the development of measures specific to this general population of children with significant mobility impairments might have resulted in outcome measures that were more sensitive to the anecdotal benefits. At this stage, we did not find evidence of improved bowel function, mood, or the provision of physical activity.

Limitations—There is a potential for bias with parent proxy questionnaires since there is a significant financial investment in purchasing or leasing the Trexo, such that parents are likely to want to see that it helps their child. We could not control for this bias; however, not reporting improvements in all areas suggested that this was not the only reason for the reported results. Another challenge was the variable timing of when parents completed the questionnaires. Several participants did not have a baseline assessment completed until one to two weeks after starting training with the Trexo, which may have diminished the amplitude of change observed. Early participants who did not have questions appeared as a forced response and thus some scores were not completed. FMS responses were unclear in the interpretation of the option “uses a wheelchair,” where several parent proxies selected this response when the child was pushed in a wheelchair by another individual (in which case the correct response would be “does not complete the distance.” In cases where this was unable to be clarified, an FMS score of 1 (“uses a wheelchair”) was excluded.

Future research should include longer-term follow-up to see if changes observed in the initial month persist and if there are any impacts, such as improvements in functional ability, that would not be reasonably expected to be seen after only four weeks. Objective measures of physical activity would add meaningful information. Additionally, an experimental rather than observational study design would support consistent Trexo use and shed light on the relationship between usage and its impacts.

## 5. Conclusions

The Trexo is an RAGT used by people aged two to 24 years old with significant mobility limitations. Training three times per week is feasible in the community, which is novel for this type of device. That 95% of participants were classified as GMFCS levels IV or V highlighted that the primary users of this device comprised a subgroup often not included in RAGT literature and who have very few options for upright mobility. Sleep disturbance, a common problem for children with neurologic impairment, improved within the initial month of using the Trexo. Future experimental studies incorporating objective outcomes over a longer training period are needed to understand any dose-response relationships and prescribe target training volumes. Until then, the Trexo is a viable means of achieving upright assisted mobility for children and young adults who otherwise have few options available.

## Figures and Tables

**Figure 1 children-10-00437-f001:**
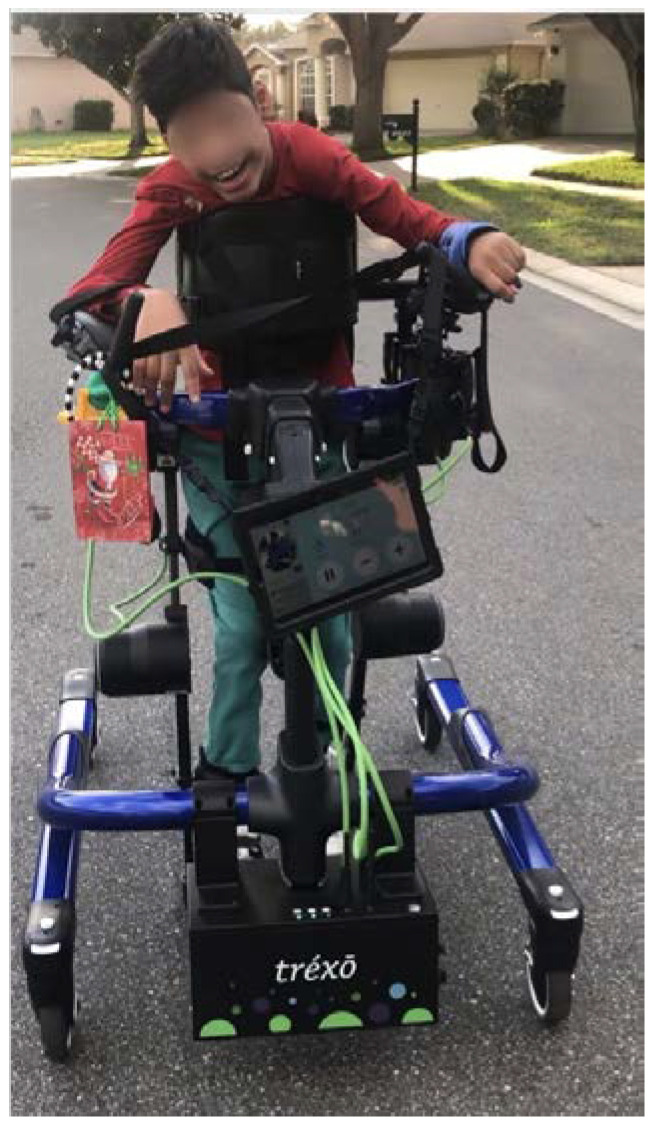
A child using the Trexo device in the community. The smile is meaningful as it represents the positive experiences reported by families. This photo was shared with permission.

**Figure 2 children-10-00437-f002:**
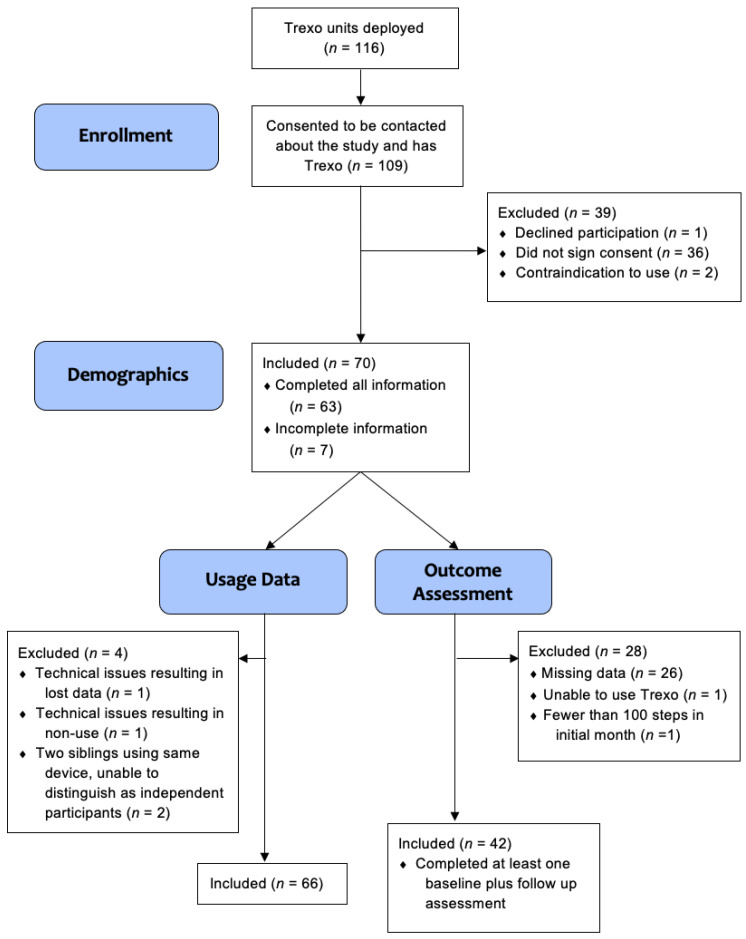
Participant flow diagram.

**Figure 3 children-10-00437-f003:**
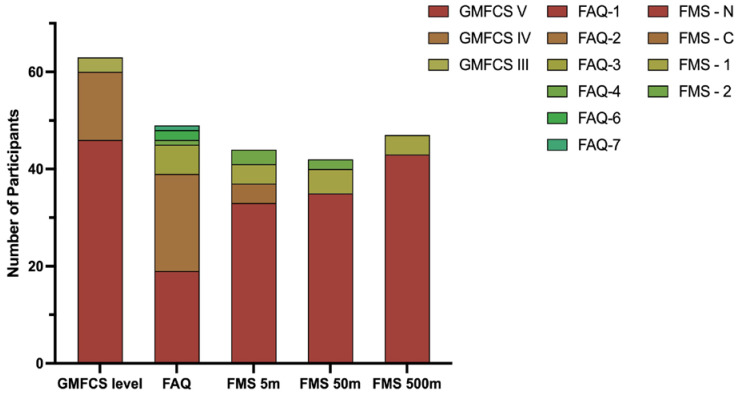
Functional mobility at baseline, including all reported levels for GMFCS, FAQ-10, and FMS.

**Figure 4 children-10-00437-f004:**
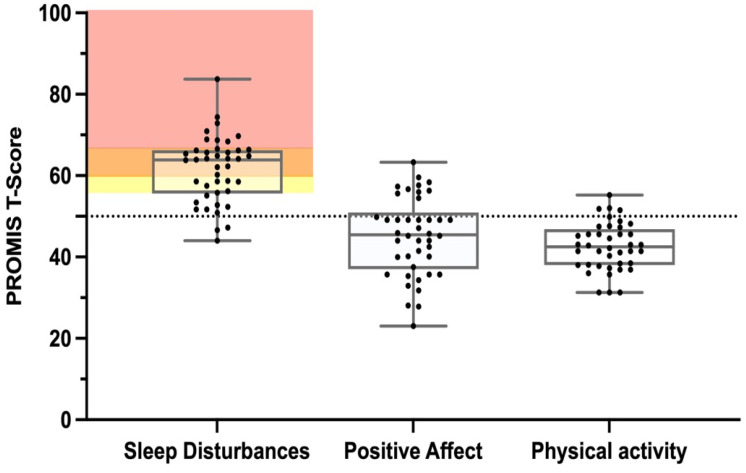
PROMIS parent proxy scores for sleep disturbance, positive affect, and physical activity before training with the Trexo. A *t*-score of 50 represents the mean of the general population. Severity of sleep disturbance thresholds are indicated by shading: red = severe (≥66), orange = moderate, and yellow = mild (<60 but ≥56).

**Figure 5 children-10-00437-f005:**
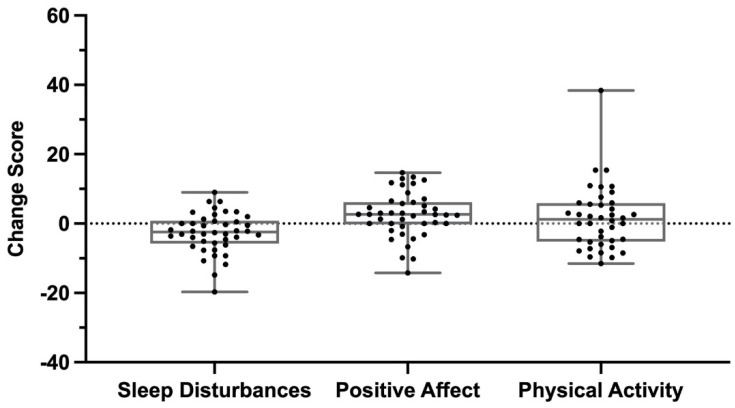
PROMIS parent proxy change scores from baseline to one month of Trexo use.

**Table 1 children-10-00437-t001:** Participant demographics.

Age		Sex		GMFCS		Diagnosis	
*n* = 70	Years	*n* = 70	*n* (%)	*n* = 63	*n* (%)	*n* = 70	*n* (%)
Median	7	Male	41 (59)	III	3 (5)	Non-progressive	58 (83)
25th–75th %	4–9	Female	29 (41)	IV	14 (22)	Progressive	5 (7)
Range	2–24			V	46 (73)	Unknown	7 (10)

## Data Availability

Anonymized data available on request.

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
