# Peer review of "Robot-Assisted Gait Training with Trexo Home: Users, Usage and Initial Impacts"

_children, 2023, doi:10.3390/children10030437_

Round 1

Reviewer 1 Report

I would like to congratulate the authors of the study for their scientific rigour, as well as for the exposition of everything related to the study.

I believe it should be considered for publication in the current format.

I add the following comments:

Novelty: The study is sufficiently original to be considered for publication in the journal.

Scope: The topic fits the scope of the journal, taking into account that it is an analysis of a possible area of action in the health sciences.

Relevance: The data are adequately interpreted, and are sufficiently scientifically sound to be relevant. The sample size is optimal and the methodology is sufficiently justified. The hypotheses of the study are sufficiently justified in the section.

Quality: The article is adequately written, well structured and supported by similar studies that raise issues to be worked on in this project. The results section is optimal and very clear in its exposition.

Scientific soundness: The study presents the necessary scientific soundness to extract highly significant and valid results for the scientific community. The sample size is clearly justified and in this justification we think that it is an adequate sample size for the methodological characteristics of the study. The methods and instruments used are clearly described, allowing the study to be replicated under similar conditions.

Readership interest: We think that the scientific robustness is sufficient to attract a significant number of readers from the scientific community.

Overall merit: Overall, we think the study could have sufficient scientific strength to be considered for publication. We encourage researchers to pursue the topic further and to increase knowledge on this important topic in the health sciences in future studies.

Level of English: I do not feel qualified to judge on English language and style.

Author Response

Thank you for your positive feedback and support for publication.

Reviewer 2 Report

I congratulate the authors for the excellent work, and I would like to help with some points that I still think can be strengthened, below are my considerations:

- The introduction needs to be better contextualized with the clinical relevance of the study and add the expected hypotheses.

- Methodology: needs to be more detailed in the variables collected on the topic: Parent-Proxy Outcomes.

- Results are well described

- Discussion: need to add one more topic about the advantages and disadvantages of the Trexo robotic gait trainer and its cost-effectiveness;

- Conclusion should be clearer and in accordance with the results found, with more details of information.

Author Response

Point 1: The introduction needs to be better contextualized with the clinical relevance of the study and add the expected hypotheses.

Thank you for the suggestion to provide more information. The introduction has been edited to improve understanding of the clinical relevance and purpose of the study including our hypothesis.

Point 2: Methodology: needs to be more detailed in the variables collected on the topic: Parent-Proxy Outcomes.

We have added details to explain why these measures were collected. Please let us know if there are other specific details that are missing.

Point 3: Results are well described.

Thank you.

Point 4: Discussion: need to add one more topic about the advantages and disadvantages of the Trexo robotic gait trainer and its cost-effectiveness.

A cost-benefit analysis needs to be done to determine cost-effectiveness, which is a larger project than we can include here, but an important suggestion for future research.

Point 5:  Conclusion should be clearer and in accordance with the results found, with more details of information.

The conclusion has been edited to include more details and relating back to the results.

Reviewer 3 Report

Introduction: Kindly provide more details about current technique/ technology and identify the gap how your study can fill that gap and clearly mention the gap of the study.

Methods: Kindly blur participant face or draw a filled circle to hide face of the participant. 

Conclusion: too short in my opinion it should address the aim of the study. 

Author Response

Point 1: Introduction: Kindly provide more details about current technique/ technology and identify the gap how your study can fill that gap and clearly mention the gap of the study.

Thank you for the suggestion to provide more information. The introduction has been edited to include more details, including the current gap this research aims to fill.

Point 2: Methods: Kindly blur participant face or draw a filled circle to hide face of the participant.

Thank you for this suggestion. We do have permission from the family to use the photo but have blurred the eyes and nose to comply with the editorial requirements. We feel leaving the smile visible is a meaningful representation of the positive experiences reported by families.

Point 3: Conclusion: too short in my opinion it should address the aim of the study. 

The conclusion has been edited to include more details.